# Correction of Accounting Errors through Post Balance Sheet Event Analysis for Romanian Companies

**Monica Laura Zlati [1], Valentin Marian Antohi [2,*]  and Petronela Cardon [2]**

[1] Department of Accounting, Audit and Finance, Stefan cel Mare University, 720229 Suceava, Romania; sorici.monica@usm.ro

[2] Departament of Business Administration, Dunarea de Jos University/Faculty of Economics and Business Administration, 800001 Galati, Romania; cardon.petronela@yahoo.com

* Correspondence: valentin_antohi@yahoo.com; Tel.: +40-731-221-001

**Abstract:** The study scope is to present the typology of the events analyzed through our research and their impact on the quality of reported financial data. The objectives of the study are to analyze the vulnerability of enterprises according to methodological criteria such as risks and calculations of the risk profile, as well as to establish the necessary measures for correcting the accounting errors based on the conclusions drawn from the analysis. The method used is prospective, financial analysis of the data taken from the financial statements of the companies included in the sample, dynamic for a period of 6 years (2011–2016). Based on the method used, a risk model has been conceptualized to identify the vulnerabilities and risks reported in the financial statements and to define a company risk profile based on which error correction measures can be adopted. Considering the amplitude of the necessary check-ups and the methodology of the imposed accounting treatments, we believe that the topic addressed is a real area of interest for the professional accountants because it organizes the application procedures and limits the impact of errors on the quality of financial reporting in Romania.

**Keywords:** accounting errors; corrections; accounting treatments; procedures; financial report; IAS-8

**JEL Classification:** M41; M21

## 1. Introduction

The changes discussed in this paper are the result of new information held by the entity, classified as error corrections. Prior periods errors are omissions or misstatements in the financial statements and are the result of non-use or misuse of information that:

- was available when those financial statements were authorized for issue
- could reasonably be taken into account in the preparation and presentation of the financial statements.

For transactions of a similar nature, the entity should use the same accounting policies.

In accounting practice, the change of an estimate is the action of adjusting a carrying amount based on new information, after the balance sheet date, in the given case. The reality of the new information is the result of the evaluation, expertise or analysis of the context that triggered the specific event, which materialized out of an authentic supporting document. Depending on the revised accounting category, the accounting estimates are classified as:

- Accounting estimates generated by changes in the legislation;
- Social estimates generated by changes in employee remuneration policies;

- Economic estimates generated by the uncertain financial situation of a third-party customer/ supplier that is in a contractual relationship with the commercial company at the time of the balance sheet.

Depending on when an accounting error occurs, the way it is dealt with at the time of the balance sheet is different, as follows:

- retrospective application by implementing new (modified) accounting policies on transactions or business conditions established but not yet completed at the balance sheet date;
- retrospective reprocessing consisting of canceling the effects of errors produced in the previous period;
- prospective application, namely recognition of the effect of changes in accounting estimates.

For a better understanding of the applicable treatments to those modifications can be seen in the Figure 1 below:

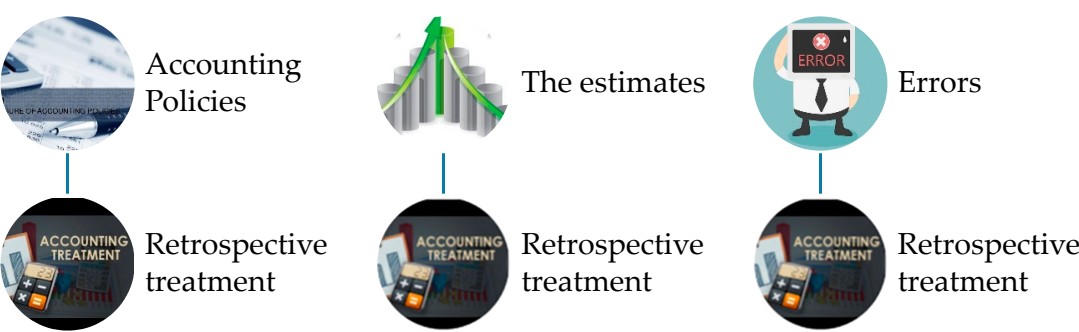

**Figure 1.** Accounting treatments in a causal relation with the main generating events.

The general rule regarding a change in accounting policies is that this change should only be applied retroactively. Another perspective, prospective or prospective application is required in cases where it is not reasonably possible to calculate the previous effect of changes in accounting policies and is permitted only if the accounting standard requires recognition in the profit and loss account and the loss of costs that were previously capitalized (Figure 2).

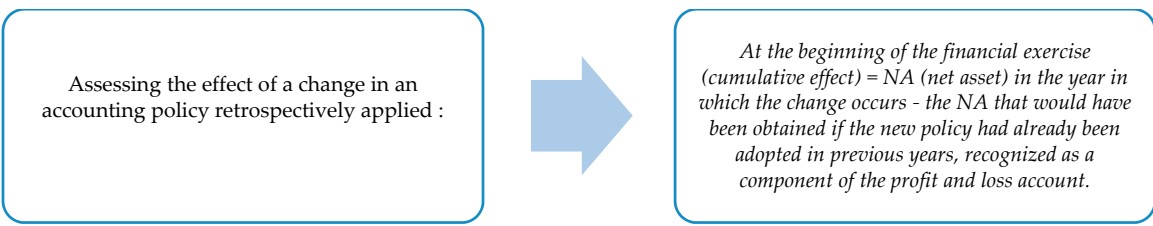

**Figure 2.** Assessment of the effect of a change in an accounting policy applied retrospectively.

It should also be emphasized that all such changes are allowed only if they are duly motivated, if they are made for a better representation of the financial statements and if the transactions are reflected through them.

Practically, IAS-8 (IASC Foundation 2018b), establishes the specific accounting policies, policies and practices of an entity for the purpose of reporting the compliant financial statements. It also defines the materiality of accounting errors by the dimensions, impact, or economic influence generated by them. IAS-8 (IAS-8.13), regulates the permanent nature of the methods as a rule for the treatment of errors. Changes in accounting policy are prerequisites (IAS-8.14) through one of two factors: either they are required by the standard or are more relevant in terms of effects on financial performance.

The accounting treatment of changes in estimates using the above methods involves the evaluation of the items in the financial statements by applying the professional judgment based on recent information.

Subsequent accounting events are dealt with in IAS 10 (Events after the Reporting Period). The current IAS 10 (International Accounting Standards Committee Foundation (IASC Foundation) 2018a) provides some limited modifications regarding: new information existing at the time of the financial statements are authorized to be published; removing the option for recognizing a liability for dividends relating to the year-end financial statement that is proposed or approved after the balance sheet date, but before it is authorized and published. Thus, an enterprise may provide the required information on such dividends, either through the balance sheet or as a separate item of equity, or in the explanatory notes; this confirms that the company must update the information on the existing situations at the balance sheet date based on any new information that occurred after that date but until the reporting date; eliminating the provision for rectifying financial statements if an event occurs after the balance sheet data indicates that the going concern principle is violated in respect of a significant portion of the reporting entity's activity. Under these circumstances, in accordance with IAS 1—Financial Statements, the business continuity principle applies to an entity as a whole or; some clarifications on examples of events involving correction and events that do not involve any adjustment; and some stylistic improvements.

An extremely important fact that can be noticed is that if an entity starts to implement a restructuring program or communicates its main aspects to the interested parties, it should report this only after the closing date of the financial year in accordance with IAS 10—Events after the end of the reference year if its restructuring is significant, and its non-inclusion could influence the economic decisions taken by users based on financial statements.

IAS 8—Accounting Policies, Changes in Accounting Estimates and Errors, which was first approved in 1976 in Exposure Draft E8—The Treatment in the Income Statement of Unusual Items and Changes in Accounting Estimates and Accounting Policies. Following successive changes, the standard was defined later, in 1995, the last review being made in 2005.

The IAS-8 is applied in the specific accounting policies, regulations and practices of their application by an entity for reporting the compliant financial statements. The standard also defines the materiality of accounting errors by their size, impact, or economic influence.

IAS-8 (IAS-8.13), regulates the consistency of accounting policies as the rule of treating errors. Changes in accounting policies (IAS-8.14) are permitted through one of two possibilities: either required by a standard or the results in the financial statements provide more relevant information in terms of the effects on the financial performance.

The accounting research performed by experts in the field is often focused on the academic feature of the accounting phenomenon, highlighting the structural changes of the accounting systems of Parker et al. (2011). Financial reporting has been structured in a compliance exercise instead of being developed as a means of innovation and experimentation to provide the best information to constituents (Dichev et al. 2012). This practice impedes on the development of entrepreneurial and managerial knowledge (Ryan 2010) as it is difficult for many leaders of economic entities to implement the methodological aspects investigated (Rutherford 2010).

This issue has generated ample debates in the specialized literature, highlighting the impact of the practical side of research, especially in the field of accounting policies (Singleton-Green 2010). Thus, mixed teams of students, professors and researchers have been established in academic centers with the purpose of increasing the impact of academic research in professional practice and of shaping entrepreneurs and managers able to use the vast already existing research (Tilt 2010). The clarity and understanding of the financial statements by the management of an enterprise is in a downward trend, as demonstrated by a study concluding that the financial statements are "unreadable" (Bonsall et al. 2017).

Other authors have discussed the issue of the utility of creating leading accounting research, given the circumstances of the ever-changing IAS and IFRS conceptual framework becoming increasingly relevant (Unerman and Brendan 2010).

As far as financial situations are concerned, there is growing interest in the specialized literature to involve managers in voluntary reporting and in developing quality financial statements (Bamber et al. 2010). Some authors (Cohen and Malloy 2011), emphasize the quality of financial statements disclosure in corporations using BIA (Business Intelligence Advisors). However, there is widespread dissatisfaction with the financial system of enterprises being reported to investors. Discontent is corroborated with empirical evidence which consistently confirms a decline in the ability of financial information to provide a demonstration of the performance of an enterprise (Lev 2018).

These aspects of quality financial statements disclosure have changing effects on accounting regulations because of the size of transactions and the volume of repatriated amounts as profits of multinational companies (Graham et al. 2011). Other authors (Gassen and Schwedler 2010), argue on the concept of reasonable quantification of the usefulness of accounting information for users of financial statements. All these steps have been aimed at correcting accounting errors and turning financial statements pieces of information into high quality information. Analyzing the last several years of legislative changes and extensive discussions on accounting issues, we can discover that these areas are affecting financial reporting systems, increasing the extent of errors among specialists (Dyer et al. 2017).

These approaches are the subject of IFRS 9—Financial Instruments (IASB 2018), as well as IAS 27—Consolidated and Separate Financial Statements (IASB 2009), given the trend towards the globalization of business. An essential step in the preparation of the financial statements is the assessment of the financial situation, with emphasis on the significance of the accounting errors discovered by the auditor or through the internal controls of the firm (Acito et al. 2018).

The need for adapting accounting systems of reference to the global business dimension has required that the International Accounting Standards Committee Foundation (IASC Foundation), later the International Accounting Standards Board (IASB) create sets of uniform rules called policies to streamline application and interpretation in the global context of International Accounting Standards, IAS 8—Accounting Policies, Changes in Accounting Estimates and Errors (IASC Foundation 2018b).

The accounting policies set out in IAS 8 represent a set of specific principles, rules, conventions and practices adopted by an entity for the presentation of qualitative financial statements and in keeping with reality (IASC Foundation 2018b).

The combination of key financial elements is the traditional way to show the performance of a business but (Lev and Gu 2016) changes in accounting estimates are in fact the true adjustment of accounting information meant for financial reporting. These estimates are sensitive to managers' interests in the strategic objectives of the entity and therefore the role of accounting policies is to be the guarantor of the unitary and exhaustive interpretation of financial information meant for financial reporting because a lack of transparency in financial statements and reports undermines trust among investors and their willingness to invest (CFA Institute 2013).

There is a special interest for researchers in the last period, in terms of accounting treatments and error correction, (the last 5 years) because more articles of interest have been published. Some authors (Advani and Malde 2018) presented the general econometric framework for the quantification of the effects of linear modeling of social effects on accounting group, through error measurement. The snowball model achieves the problem-handling phenomenon by estimating unbiased parameters, with the authors believing that the results after error handling become more feasible and provide the convenience of handling the treated phenomenon.

In the article *Imperfect accounting and Reporting Bias* (Fang et al. 2017), there is a direct and strong connection between errors and injuries. This connection is demonstrated by statistical methods, the authors pointing out that the more detectable the damage is, more significant the error and the future earnings of the firm are. The results of the study highlight the accounting imperfections in relation of top leadership incentives and the imperfections in understanding the provisions of the

accounting standards. The proposed model highlights the relationship between the cost of errors and the endogenous variables of the reporting effort.

To estimate the impact of intentional violation of accounting standards (GAAP study), (Zakolyukina 2018) developed a dynamic model for estimating accounting errors at the CEO level with the intention of affecting the stock market price over a 5 years horizon, in which case an average of 13 errors occurred in 91% of cases, which is a significant threshold for the study case presented. Modeling values have shown that the company size influences the analyzed indicator.

Approaches to the quality of accounting information and systemic risk were addressed through a regression model based on the least squares method (Xing and Yan 2019), through which it has been demonstrated (by studying a computerized database) and realized over a period of 50 years (1962–2012), which increased the quality of accounting information and is a direct cause of systemic risk reduction.

The purpose of this paper is to provide additional solutions to improve the quality of information in the financial statements in connections with post-balance sheet events IAS 10 (International Accounting Standards Committee Foundation (IASC Foundation) 2018a) or adjustments resulting from changes in accounting estimates or errors IAS 8 by applying an economic model that highlights the balance sheet items that require adjustments, as well as the impact of these adjustments on the financial position and result. Thus, the conditions that should be met in order for the corrections made to be appropriate with the information founded in the financial statements, following the events that take place after the closing of the balance, correlated with the retrospective treatments applied in case of changes in accounting policies, or the correction of the errors and the prospective treatments, in the case of accounting estimates.

## 2. Methodology

The study was carried out on a sample of 40 companies, the most profitable companies in Galati, Romania, covering the main branches of economic activity in the area, the results obtained being expanded at the national and international level.

The area under analysis is characterized by the following significant aspects for the study:

- an industrialized zone with access to the Danube (port city), predominantly steel, naval, Agri-food sector, and construction industries;
- the zone is in a process of reorganization, intensifying trade based on access to the port area;
- the zone is in a pole of urban concentration, being the second most urbanized location in Romania after Bucharest, with high investment potential;

This study is based on financial data reported by the top 40 companies (Figure 3) in Galati county, Romania, ranked in terms of a 6-year ($i = \overline{1,6}$), turnover (2011–2016).

The research aimed at studying the dynamics of profitability indicators (gross profit, net profit), the revenue/expenses ratio and the structure indicators of balance sheet assets and liabilities and their relation to ownership equity, net profit and turnover.

| Crt no. | Companies | Turnover mil.euro 2016 | Gross profit mil.euro 2016 | Net income mil.euro 2016 | Provisions mil.euro 2016 | Equity mil.euro 2016 |
|---|---|---|---|---|---|---|
| 1 | ARCELORMITTAL GALAȚI SA | 763.45 | -59.76 | -59.76 | 38.43 | 268.70 |
| 2 | ARABESQUE SRL | 348.99 | 13.60 | 11.04 | 0.00 | 131.52 |
| 3 | MAIRON GALATI SA | 180.40 | 10.93 | 9.46 | 0.00 | 65.51 |
| 4 | PRUTUL SA | 165.60 | -5.85 | -5.85 | 0.00 | 5.87 |
| 5 | SANTIERUL NAVAL DAMEN GALATI SA | 120.76 | 2.05 | 0.24 | 0.34 | 67.59 |
| 6 | TANCRAD SRL | 73.48 | 1.77 | 1.48 | 0.00 | 25.36 |
| 7 | ARCADA COMPANY SA | 65.40 | 10.05 | 8.78 | 14.83 | 62.83 |
| 8 | COMPANIA DE NAVIGATIE FLUVIALA ROMANA NAVROM SA | 45.65 | 1.45 | 1.28 | 0.84 | 70.09 |
| 9 | NEXT ENERGY PARTNERS SRL | 45.47 | 0.95 | 0.81 | 0.00 | 0.92 |
| 10 | BAUROM CONSTRUCT SRL | 39.62 | 1.58 | 1.33 | 0.00 | 8.30 |
| 11 | ALEWIJNSE MARINE GALAȚI SOCIETATE PE ACȚIUNI | 36.97 | 2.46 | 2.26 | 0.19 | 6.25 |
| 12 | ANGHEL N.G. SRL | 34.73 | 0.14 | 0.11 | 0.00 | 0.74 |
| 13 | MAIRON TUBES S.R.L. | 28.67 | 0.65 | 0.57 | 0.00 | 8.62 |
| 14 | DMT MARINE EQUIPMENT SOCIETATE PE ACȚIUNI | 27.55 | 2.43 | 2.06 | 0.33 | 8.22 |
| 15 | DYNAMIC SELLING GROUP SRL | 25.10 | 1.53 | 1.31 | 0.00 | 3.50 |
| 16 | PHOENIX SLAG SERVICES SRL | 23.08 | 6.43 | 5.39 | 0.23 | 25.99 |
| 17 | MYOSOTIS SRL | 22.95 | 0.88 | 0.67 | 0.00 | 2.62 |
| 18 | MYOSOTIS FARM SRL | 22.95 | 0.88 | 0.67 | 0.00 | 2.62 |
| 19 | CRICONS SRL | 22.31 | 0.54 | 0.47 | 0.00 | 0.95 |
| 20 | FIERCTC SIBEL SRL | 22.13 | 1.79 | 1.50 | 0.00 | 6.88 |
| 21 | CITY GAS S.R.L. | 21.82 | 0.04 | 0.03 | 0.00 | 0.82 |
| 22 | EUXIN SRL | 17.74 | 0.06 | 0.05 | 0.00 | 0.41 |
| 23 | CHORUS MARKETING AND DISTRIBUTION SRL | 16.73 | 0.20 | 0.18 | 0.00 | 0.47 |
| 24 | CARUL CU BERE 95 | 16.47 | 0.80 | 0.68 | 0.00 | 0.71 |
| 25 | VEGA 93 SRL | 16.46 | -4.34 | -4.34 | 0.00 | -13.62 |
| 26 | STAER INTERNATIONAL SA | 16.31 | 1.84 | 1.56 | 0.00 | 4.58 |
| 27 | DOLADELA COMPANY S.R.L. | 14.10 | 1.38 | 1.16 | 0.00 | 5.57 |
| 28 | S.F.TEX SA | 12.87 | 0.57 | 0.48 | 0.00 | 2.03 |
| 29 | ANDREEAS 95 EXIM SRL | 10.93 | 1.42 | 1.24 | 0.00 | 2.40 |
| 30 | ANDRADA SRL | 10.59 | 0.74 | 0.64 | 0.00 | 1.88 |
| 31 | ROJEVAS 2000 SRL | 10.04 | 1.53 | 1.31 | 0.00 | 4.91 |
| 32 | METALTRADE INTERNATIONAL SRL | 10.03 | -1.67 | -1.67 | 0.00 | 39.96 |
| 33 | NADICER SRL | 8.66 | 0.74 | 0.64 | 0.00 | 0.86 |
| 34 | ALMERA INTERNATIONAL SRL | 8.40 | 0.10 | 0.09 | 0.00 | 4.49 |
| 35 | DERPAN SRL | 8.25 | 0.73 | 0.61 | 0.00 | 2.13 |
| 36 | BAUROM SRL | 8.20 | 0.13 | 0.10 | 0.00 | 0.27 |
| 37 | RACRICOM S.R.L. | 8.05 | 0.24 | 0.21 | 0.00 | 0.71 |
| 38 | SALTEMPO SRL | 7.69 | 0.65 | 0.57 | 0.00 | 0.90 |
| 39 | NOY BUSINESS TRANZACTIONS SRL | 7.32 | 0.19 | 0.16 | 0.00 | 4.60 |
| 40 | BULROM PETROLEUM SRL | 6.99 | 0.28 | 0.24 | 0.00 | 1.01 |

**Figure 3.** Histogram ranking of Companies in Galati County, Romania, based on their 2016 turnover.

The indicators analyzed in the dynamics are those presented in Figure 4:

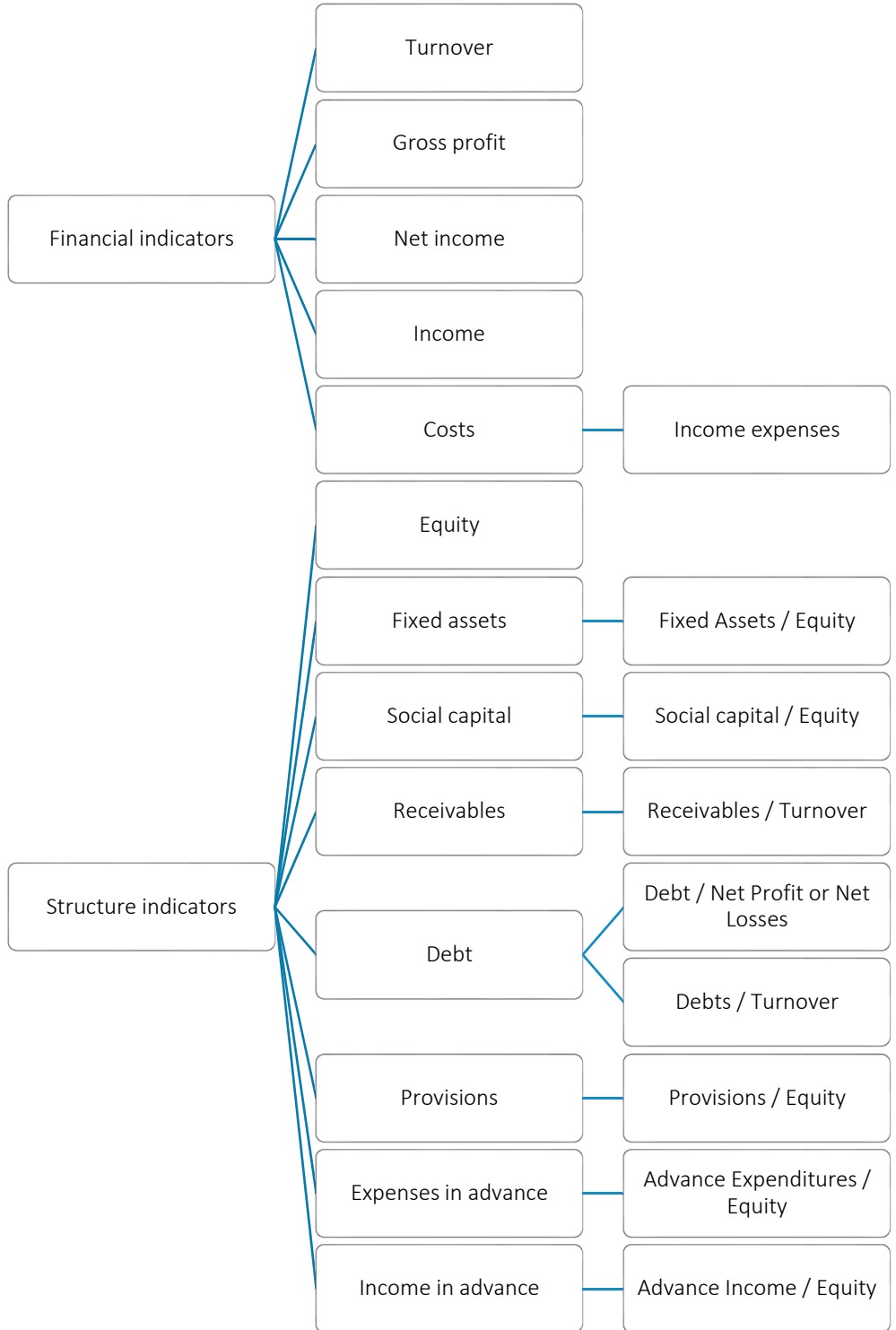

**Figure 4.** Indicators using for modeling the accounting treatments of post balance sheet events.

Out of the 40 companies that were considered, only 7 met the modeling conditions, with provisions, incomes received in advance and expenses paid in advance at the time of the balance sheet (Figure 5):

| Crt no. | Companies | Turnover mil.euro 2016 | Gross profit mil.euro 2016 | Net income mil.euro 2016 | Provisions mil.euro 2016 | Equity mil.euro 2016 |
|---|---|---|---|---|---|---|
| 1 | ARCELORMITTAL GALAȚI SA | 763.45 | -59.76 | -59.76 | 38.43 | 268.70 |
| 5 | SANTIERUL NAVAL DAMEN GALATI SA | 120.76 | 2.05 | 0.24 | 0.34 | 67.59 |
| 7 | ARCADA COMPANY SA | 65.40 | 10.05 | 8.78 | 14.83 | 62.83 |
| 8 | COMPANIA DE NAVIGATIE FLUVIALA ROMANA NAVROM SA | 45.65 | 1.45 | 1.28 | 0.84 | 70.09 |
| 11 | ALEWIJNSE MARINE GALAȚI SOCIETATE PE ACȚIUNI | 36.97 | 2.46 | 2.26 | 0.19 | 6.25 |
| 14 | DMT MARINE EQUIPMENT SOCIETATE PE ACȚIUNI | 27.55 | 2.43 | 2.06 | 0.33 | 8.22 |
| 16 | PHOENIX SLAG SERVICES SRL | 23.08 | 6.43 | 5.39 | 0.23 | 25.99 |

**Figure 5.** Companies at potential risk in terms of the post balance sheet events.

To conceptualize the proposed model, the following assumptions are made:

**Hypothesis 1 (H1).** *The risk associated with some indicators is even greater as the volatility of the cumulative financial information, relative to its own capital, is higher.*

**Hypothesis 2 (H2).** *The items reflected in the financial asset can corrected by applying relevant adjustments after an impact analysis of their dematerialization.*

**Hypothesis 3 (H3).** *The elements reflected in the financial liability are more volatile and can be corrected by applying adjustments to the type of provisions only after an impact analysis of the relationship—Volatility— Macroeconomic Volatility.*

**Hypothesis 4 (H4).** *Appropriate accounting errors are generators of stability and growth, by increasing the confidentiality of users of financial statements in the context of integrating all adjustments generated by the model.*

The model used is one of the relative series, compared by their dynamics ($n/n-1$) based on the indicators mentioned in Figure 4, from which the following indicators set out in Table 1 were extracted as key risk and adjustment enabler indicators, also comprising the risk condition and the level of impact on the global financial situation:

**Table 1.** Risk indicators.

| INDICATORS | Symbol | Non-Risk Level | Impact of the Indicator over the Total Risk |
|---|---|---|---|
| Income expenses | $\Delta_{IE_i}$ | >1 | 2 |
| Equity | $\Delta_{EQ_i}$ | >1 | 1 |
| Fixed Assets/Equity | $\Delta_{FAE_i}$ | <1 | 4 |
| Social capital/Equity | $\Delta_{SCE_i}$ | <1 | 3 |
| Receivables/Turnover | $\Delta_{RT_i}$ | <1 | 2 |
| Debts/Turnover | $\Delta_{DT_i}$ | <1 | 3 |
| Provisions/Equity | $\Delta_{PVE_i}$ | <1 | 4 |
| Advance Expenditures/Equity | $\Delta_{AEE_i}$ | <1 | 3 |
| Advance Income/Equity | $\Delta_{AIE_i}$ | <1 | 3 |

The indicators in Table 1 are further analyzed in detail below:

The submitted model analyses the indicators in their dynamics according to the formulas below:

$$\textbf{Income expenses—}\Delta_{IE_i} = \frac{IE_i}{IE_{i-1}} = \frac{\Delta_{I_i}}{\Delta_{C_i}} = \frac{\frac{I_i}{I_{i-1}}}{\frac{E_i}{E_{i-1}}}, \ i = \overline{1,6},$$

where:

- *IE*, represent Income expenses;
- *I*, represent Total Incomes presented in the balance;
- *E*, represent Total expenses presented in the balance;
- *i*, reference year and *i* − 1, previous year;

$$\textbf{Equity—}\Delta_{EQ_i} = \frac{EQ_i}{EQ_{i-1}}, \ i = \overline{1,6},$$

where:

- *EQ*, represent Equity presented in the balance;
- *i*, reference year and *i* − 1, previous year;

$$\textbf{Fixed Assets/Equity—}\Delta_{FAE_i} = \frac{FAE_i}{FAE_{i-1}} = \frac{\Delta_{FA_i}}{\Delta_{EQ_i}} = \frac{\frac{FA_i}{FA_{i-1}}}{\frac{EQ_i}{EQ_{i-1}}}, \ i = \overline{1,6},$$

where:

- *FAE*, represent Fixed Assets/Equity;
- *FA*, represent Fixed Assets presented in the balance;
- *EQ*, represent Equity presented in the balance;
- *i*, reference year and *i* − 1, previous year;

$$\textbf{Social capital/Equity—}\Delta_{SCE_i} = \frac{SCE_i}{SCE_{i-1}} = \frac{\Delta_{SC_i}}{\Delta_{EQ_i}} = \frac{\frac{SC_i}{SC_{i-1}}}{\frac{EQ_i}{EQ_{i-1}}}, \ i = \overline{1,6},$$

where:

- *SCE*, represent Social capital/Equity;
- *SC*, represent Social capital presented in the balance;
- *EQ*, represent Equity presented in the balance;
- *i*, reference year and *i* − 1, previous year;

$$\textbf{Receivables/Turnover—}\Delta_{RT_i} = \frac{RT_i}{RT_{i-1}} = \frac{\Delta_{R_i}}{\Delta_{T_i}} = \frac{\frac{R_i}{R_{i-1}}}{\frac{T_i}{T_{i-1}}}, \ i = \overline{1,6},$$

where:

- *RT*, represent Receivables/Turnover;
- *R*, represent Receivables presented in the balance;
- *T*, represent Turnover presented in the balance;
- *i*, reference year and *i* − 1, previous year;

$$\textbf{Debts/Turnover—}\Delta_{DT_i} = \frac{DT_i}{DT_{i-1}} = \frac{\Delta_{D_i}}{\Delta_{T_i}} = \frac{\frac{D_i}{D_{i-1}}}{\frac{T_i}{T_{i-1}}}, \ i = \overline{1,6},$$

where:

- *DT*, represent Debts/Turnover;
- *D*, represent Debts presented in the balance;
- *T*, represent Turnover presented in the balance;
- *i*, reference year and $i - 1$, previous year;

$$\textbf{Provisions/Equity}—\Delta_{PVE_i} = \frac{PVE_i}{PVE_{i-1}} = \frac{\Delta_{PV_i}}{\Delta_{EQ_i}} = \frac{\frac{PV_i}{PV_{i-1}}}{\frac{EQ_i}{EQ_{i-1}}}, \ i = \overline{1,6},$$

where:

- *PVE*, represent Provisions/Equity;
- *PV*, represent Provisions presented in the balance;
- *EQ*, represent Equity presented in the balance;
- *i*, reference year and $i - 1$, previous year;

$$\textbf{Advance Expenditures/Equity}—\Delta_{AEE_i} = \frac{AEE_i}{AEE_{i-1}} = \frac{\Delta_{AE_i}}{\Delta_{EQ_i}} = \frac{\frac{AE_i}{AE_{i-1}}}{\frac{EQ_i}{EQ_{i-1}}}, \ i = \overline{1,6},$$

where:

- *AEE*, represent Advance Expenditures/Equity;
- *AE*, represent Advance Expenditures presented in the balance;
- *EQ*, represent Equity presented in the balance;
- *i*, reference year and $i - 1$, previous year;

$$\textbf{Advance Income/Equity}—\Delta_{AIE_i} = \frac{AIE_i}{AIE_{i-1}} = \frac{\Delta_{AI_i}}{\Delta_{EQ_i}} = \frac{\frac{AI_i}{AI_{i-1}}}{\frac{EQ_i}{EQ_{i-1}}}, \ i = \overline{1,6},$$

where:

- *AIE*, represent Advance Income/Equity;
- *AE*, represent Advance Income presented in the balance;
- *EQ*, represent Equity presented in the balance;
- *i*, reference year and $i - 1$, previous year;

The indicators determined in this way are compared with the risk threshold according to the formula:

$$\Delta_{K_i} - R_K > 0,$$

where:

- $\Delta_{K_i}$, represent indicator level for one specific period;
- $R_K$, represent non-risk level for K indicator;
- *i*, reference year;

For values bigger than 0, the impact coefficient in Table 1 is applied.

The impact coefficient from Table 1 is applied to the risk indicators calculated based on the formula: $\Delta_{K_i} - R_K > 0$, being calculated the annual average of positive risks, which is applied as a threshold for determining gross adjustments, as follows:

$$\frac{\Delta_{K_i} * c_i}{\left( \frac{\sum_{K=1}^{9} \Delta_{K_i} * c_i}{9} \right)} > 1,$$

where : $\Delta_{K_i} * c_i$, represent indicator under the risk, pondered with an impact coefficient.

$$Net\ income_i \xrightarrow{Income\ expenses\ with\ positive\ evaluated\ risk\Delta_{IE_i} - R_{IE_i} > 0} adjusted\ Net\ income = \left(1 - \Delta_{IE_i}\right) * Net\ income_i$$

where:

- *IE*, represent Income expenses;
- *R*, represent the impact coefficient for Income expenses under the risk limit;

$$Net\ income_i \xrightarrow{Equity\ with\ positive\ evaluated\ risk\Delta_{EQ_i} - R_{EQ_i} > 0} adjusted\ Net\ income_i = \left(1 - \Delta_{EQ_i}\right) * Net\ income_i$$

where:

- *EQ*, represent Equity;
- *R*, represent the impact coefficient for Equity under the risk limit;

$$Reserves\ from\ reevaluation\ of\ fixed\ assets_i \xrightarrow{Fixed\ Assets/Equity\ with\ positive\ evaluated\ risk\Delta_{FAE_i} - R_{FAE_i} > 0} adjusted\ Reserves\ from\ reevaluation\ of\ Fixed\ Assets_i$$
$$\left(1 - \Delta_{FAE_i}\right) * Fixed\ Accets_i$$

where:

- *FAE*, represent Fixed Assets/Equity;
- *R*, represent the impact coefficient for Fixed Assets/Equity under the risk limit;

$$Provisions\ for\ insolvence\ risk_i \xrightarrow{Social\ capital/Equity\ with\ positive\ evaluated\ risk\ \Delta_{SCE_i} - R_{SCE_i} > 0} adjusted\ Provisions_i = \left(1 - \Delta_{SCE_i}\right) * Provisions_i,$$

where:

- *SCE*, represent Social capital/Equity;
- *R*, represent the impact coefficient for Social capital/Equity under the risk limit;

$$Provisions\ for\ incert\ receivables_i \xrightarrow{Receivables\ Turnover\ with\ positive\ evaluated\ risk\ \Delta_{RT_i} - R_{RT_i} > 0} adjusted\ Provisions_i = \left(1 - \Delta_{RT_i}\right) * Provisions_i,$$

where:

- *RT*, represent Receivables/Turnover;
- *R*, represent the impact coefficient for Receivables/Turnover under the risk limit;

$$Provisions\ for\ incert\ debts_i \xrightarrow{Debts/Turnover\ with\ positive\ evaluated\ risk\Delta_{DT_i} - R_{DT_i} > 0} adjusted\ Provisions_i = \left(1 - \Delta_{DT_i}\right) * Provisions_i,$$

where:

- *DT*, represent Debts/Turnover;
- *R*, represent the impact coefficient for Debts/Turnover under the risk limit;

$$Provisions_i \xrightarrow{Provisions/Equity\ with\ positive\ evaluated\ risk\Delta_{PVE_i} - R_{PVE_i} > 0} adjusted\ Provisions_i = \left(1 - \Delta_{PVE_i}\right) * Provisions_i,$$

where:

- *PVE*, represent Provisions/Equity;
- *R*, represent the impact coefficient for Provisions/Equity under the risk limit;

$$\textit{Advance Expenditures / Equity with positive evaluated risk} \Delta_{AEE_i} - R_{AEE_i} > 0$$
$$\xRightarrow{\textit{Provisions for advance expenditures}_i} \textit{adjusted Provisions}_i = \left(1 - \Delta_{AEE_i}\right) * \textit{Provisions}_i,$$

where:

- *AEE*, represent Advance Expenditures/Equity;
- *R*, represent the impact coefficient for Advance Expenditures/Equity under the risk limit;

$$\textit{Advance Income / Equity with positive evaluated risk} \Delta_{AIE_i} - R_{AIE_i} > 0$$
$$\xRightarrow{\textit{Provisions for advance income}_i} \textit{adjusted Provisions}_i = \left(1 - \Delta_{AIE_i}\right) * \textit{Provisions}_i,$$

where:

- *AIE*, represent Advance Income/Equity;
- *R*, represent the impact coefficient for Advance Income/Equity under the risk limit;

The indicators presented can be inserted into a cumulative risk model whose design is presented below, as follows:

| Indicators | Limitation | Imposed accounting treatment after balance sheet |
|---|---|---|
| Income expenses Equity | $\left\{ \dfrac{\Delta_{K_i} * c_i}{\left( \dfrac{\sum_{K=1}^{9} \Delta_{K_i} * c_i}{9} \right)} > 1 \right\}$ | Net income |
| Fixed Assets/Equity | $\left\{ \dfrac{\Delta_{K_i} * c_i}{\left( \dfrac{\sum_{K=1}^{9} \Delta_{K_i} * c_i}{9} \right)} > 1 \right\}$ | Reserves from revaluation of fixed assets |
| Social capital/Equity | | Provisions |
| Receivables/Turnover | | Provisions for doubtful receivables |
| Debts/Turnover | $\dfrac{\Delta_{K_i} * c_i}{\left( \dfrac{\sum_{K=1}^{9} \Delta_{K_i} * c_i}{9} \right)} > 1$ | Provisions for doubtful debts |
| Provisions/Equity | | Provisions |
| Advance Expenditures/Equity | | Provisions for expenses in advance |
| Advance Income/Equity | | Unsettled transactions |

The model can be applied to other companies as well, not just to the seven companies analyzed. Of the total sample of 40 companies analyzed, only 7 were selected based on non-zero provisions (Figure 5).

## 3. Results

Of the 7 companies analyzed, ARCELOR MITTAL GALATI SA was selected for modeling, being the largest company in the group of studied companies. The gross values that will be adjusted in relative series according the developed model are based on the information presented in the financial situations of the past 6 years, in keeping with the analyzed indicators.

For the company ARCELOR MITTAL GALATI SA, the dynamics of the indicators analyzed are presented in Table 2, based on annual growth rates:

**Table 2.** Dynamics of the indicators for ARCELOR MITTAL GALATI SA.

| ARCELOR MITTAL GALATI SA | 2016/2015 | 2015/2014 | 2014/2013 | 2013/2012 | 2012/2011 |
|---|---|---|---|---|---|
| Turnover | 91.15% | 105.89% | 96.73% | 68.53% | 86.06% |
| Gross profit | 91.18% | 61.72% | 65.02% | 320.97% | 268.38% |
| Net income | 91.18% | 61.72% | 65.02% | 320.97% | 268.38% |
| Income | 90.63% | 104.71% | 94.39% | 73.21% | 95.80% |
| Costs | 90.67% | 99.72% | 89.68% | 83.54% | 103.00% |
| Income expenses | 99.96% | 105.01% | 105.24% | 87.64% | 93.01% |
| Equity | 81.81% | 83.40% | 83.26% | 72.73% | 107.26% |
| Fixed assets | 95.51% | 101.74% | 90.20% | 81.08% | 113.51% |
| Fixed Assets/Equity | 116.75% | 121.99% | 108.34% | 111.47% | 105.83% |
| Social capital | 100.00% | 100.00% | 100.00% | 100.00% | 100.00% |
| Social capital/Equity | 122.24% | 119.90% | 120.10% | 137.49% | 93.23% |
| Receivables | 113.63% | 65.23% | 136.79% | 97.25% | 178.91% |
| Receivables/Turnover | 124.66% | 61.61% | 141.40% | 141.92% | 207.89% |
| Debt | 130.59% | 89.33% | 104.64% | 114.25% | 138.32% |
| Debt/Net Profit or Net Losses | 143.23% | 144.75% | 160.93% | 35.59% | 51.54% |
| Debts/Turnover | 143.27% | 84.37% | 108.17% | 166.71% | 160.73% |
| Provisions | 99.82% | 118.24% | 101.77% | 87.29% | 97.70% |
| Provisions/Equity | 122.02% | 141.77% | 122.23% | 120.02% | 91.08% |
| Expenses in advance | 239.11% | 97.82% | 85.18% | 90.90% | 110.96% |
| Advance Expenditures/Equity | 292.28% | 117.29% | 102.31% | 124.98% | 103.45% |
| Income in advance | 91.18% | 61.72% | 65.02% | 320.97% | 268.38% |
| Advance Income/Equity | 78.09% | 50.59% | 60.02% | 287.95% | 253.60% |

Following the dynamics presented in Table 2, based on the harmonization indices in Table 1, the model seeks to establish the following risks for ARCELOR MITTAL GALATI SA (Table 3):

**Table 3.** The risk of post balance sheet events reflected through the modelled indicators within ARCELOR MITTAL GALATI SA.

| ARCELOR MITTAL GALATI SA | 2016/2015 Risk | 2015/2014 Risk | 2014/2013 Risk | 2013/2012 Risk | 2012/2011 Risk |
|---|---|---|---|---|---|
| Income expenses | 0.09% | 0.00% | 0.00% | 24.73% | 13.97% |
| Equity | 18.19% | 16.60% | 16.74% | 27.27% | 0.00% |
| Fixed Assets/Equity | 67.02% | 87.97% | 33.35% | 45.88% | 23.30% |
| Social capital/Equity | 66.72% | 59.71% | 60.31% | 112.46% | 0.00% |
| Receivables/Turnover | 49.32% | 0.00% | 82.81% | 83.83% | 215.78% |
| Debts/Turnover | 129.82% | 0.00% | 24.52% | 200.13% | 182.19% |
| Provisions/Equity | 88.09% | 167.09% | 88.91% | 80.08% | 0.00% |
| Advance Expenditures/Equity | 576.85% | 51.86% | 6.92% | 74.95% | 10.34% |
| Advance Income/Equity | 0.00% | 0.00% | 0.00% | 563.84% | 460.81% |
| Average | 110.68% | 42.58% | 34.84% | 134.80% | 100.71% |

Based on applying the model, the necessary adjustments to be made in three chapters of the balance sheet are (Table 4):

- net profit/loss;
- reserve din from revaluation of fixed assets;
- provision adjustments.

It can be noticed that the post balance sheet events (Figure 6) have a negative impact on own capitals, namely that they are diminishing, based on the risks associated with the doubtful receivables of the accrued expenses or deferred revenues expected in 2012 and 2013.

**Table 4.** The value of adjustments after applying the model (Euro).

| Imposed Accounting Treatment after Balance Sheet | 2016 | 2015 | 2014 | 2013 | 2012 |
|---|---|---|---|---|---|
| Adjusted reserve | 0 | 11,338,787 | 0 | 0 | 0 |
| Adjusted provision | 30,312,479 | 37,774,873 | 40,564,809 | 19,633,844 | 3,544,289 |
| Adjusted equity | 238,387,244 | 302,019,726 | 353,259,782 | 453,362,478 | 646,766,139 |
| Of which net income | −67,870,809 | −66,259,167 | −98,184,538 | −175,678,604 | −83,986,054 |
| Adjusted net income | −8,114,720 | −720,760 | 8,007,878 | −12,356,984 | −33,102,620 |

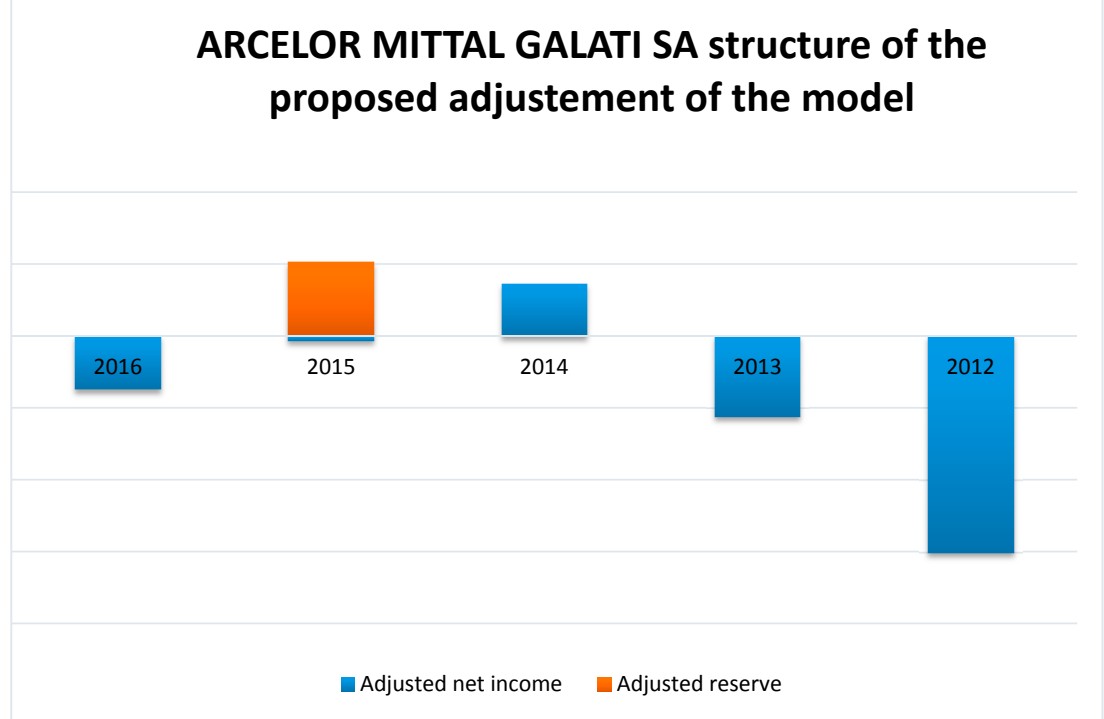

**Figure 6.** The dynamics of the adjustments proposed based on the post balance sheet events model at ARCELOR MITTAL GALATI SA.

Applying the model to the seven companies that use provisions through the adopted accounting policies demonstrates that there is a difference between the reported amount of equity and its actual value, as follows:

1. **ARCELOR MITTAL GALATI SA** There is a decrease in the analyzed range of the shareholders' equity stake based on the necessary adjustments of the established provisions and the declared net profit. Thus, the company accounted for a bad will at the end of 2016 based on the negative net result and the decrease in fixed assets and current assets by 40% compared to 2011 (Figure 7).

2. **SANTIERUL NAVAL DAMEN SA GALATI** There is an increase in the analyzed range of shareholders' equity stake based on the adjustments to reserves from the revaluation, the provisions and the net income. Thus, at the end of 2016, the company accounted for a goodwill based on the positive net result and the increase in fixed assets and current assets by 31% compared to 2011 (Figure 8).

3. **ARCADA COMPANY SA GALATI** There is an increase in the analyzed range of shareholders' equity stake, which, following the use of the model, has turned out to be surrealistic based on provision adjustments and net income. Thus, at the end of 2016, the company accounted for a goodwill based on the positive net result and the increase in fixed assets and current assets by 213% compared to 2011 (Figure 9).

4.  **COMPANIA DE NAVIGATIE FLUVIALA ROMANA NAVROM SA GALATI** There is an increase in the analyzed range of shareholders' equity stake, which, following the use of the model, has turned out to be insufficiently assessed (with the inflexion point in 2015) based on provision adjustments and net income. Thus, at the end of 2016, the company accounted for a goodwill based on the positive net result and the increase in fixed assets and current assets by 102% compared to 2011 (Figure 10).

5.  **ALEWIJNSE MARINE SA GALATI** There is an increase in the analyzed shareholders' equity stake, which, following the use of the model, proved to be realistic, the Equity and Adjusted Equity trend equations being approximately equal on the assessed range. The increase is based on negative reserve adjustments and positive provisions adjustments. Thus, at the end of 2016, the company accounted for a goodwill based on the positive net result and the increase in fixed assets and current assets by 158% compared to 2011 (Figure 11).

6.  **DMT MARINE EQUIPMENT SA GALATI** There is an increase in the analyzed range of shareholders' equity stake, which, following the use of the model, has turned out to be surrealistic based on provision adjustments and net income. Thus, at the end of 2016, the company accounted for a goodwill based on the positive net result and the increase in fixed assets and current assets by 213% compared to 2011 (Figure 12).

7.  **PHOENIX SLAG SERVICES SRL GALATI** There is an increase in the analyzed range of shareholders' equity stake, which, following the use of the model, has turned out to be surrealistic. The increase is based on negative reserves adjustments and positive provision adjustments and net income. Thus, at the end of 2016, the company accounted for a goodwill based on the positive net result and the increase in fixed assets and current assets by 344% compared to 2011 (Figure 13).

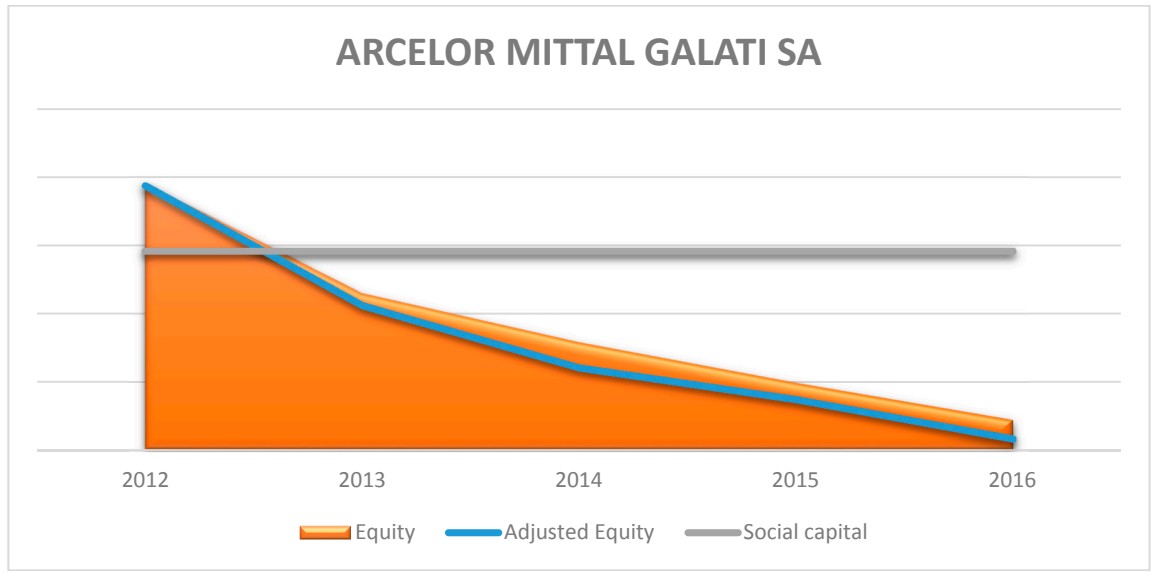

**Figure 7.** Dilution of social capital following post balance sheet events adjustments for ARCELOR MITTAL GALATI SA.

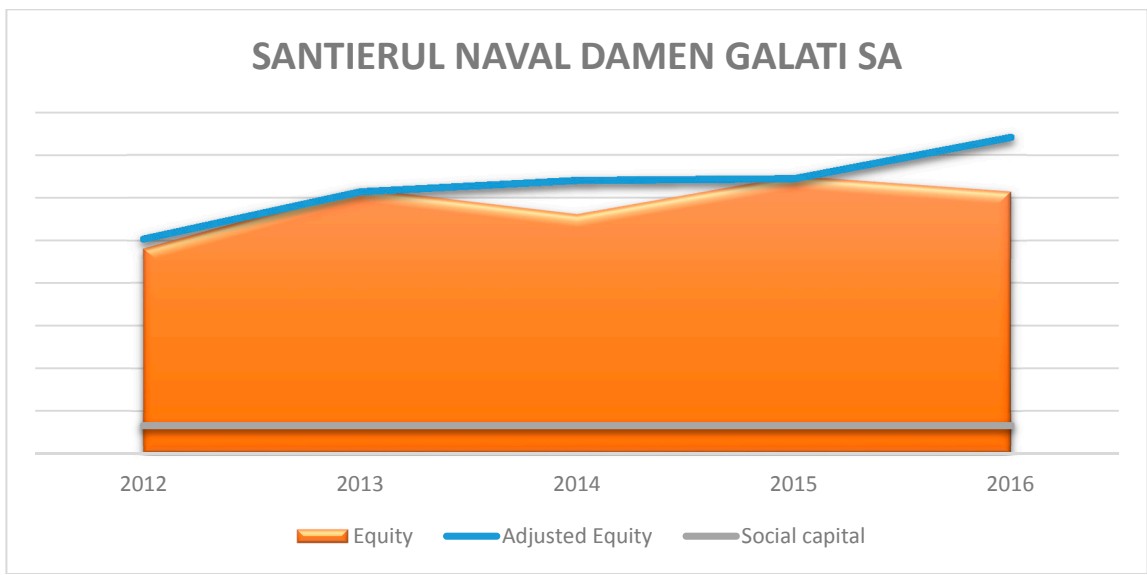

**Figure 8.** The increase in social capital following post balance sheet events adjustments for SANTIERUL NAVAL DAMEN SA GALATI.

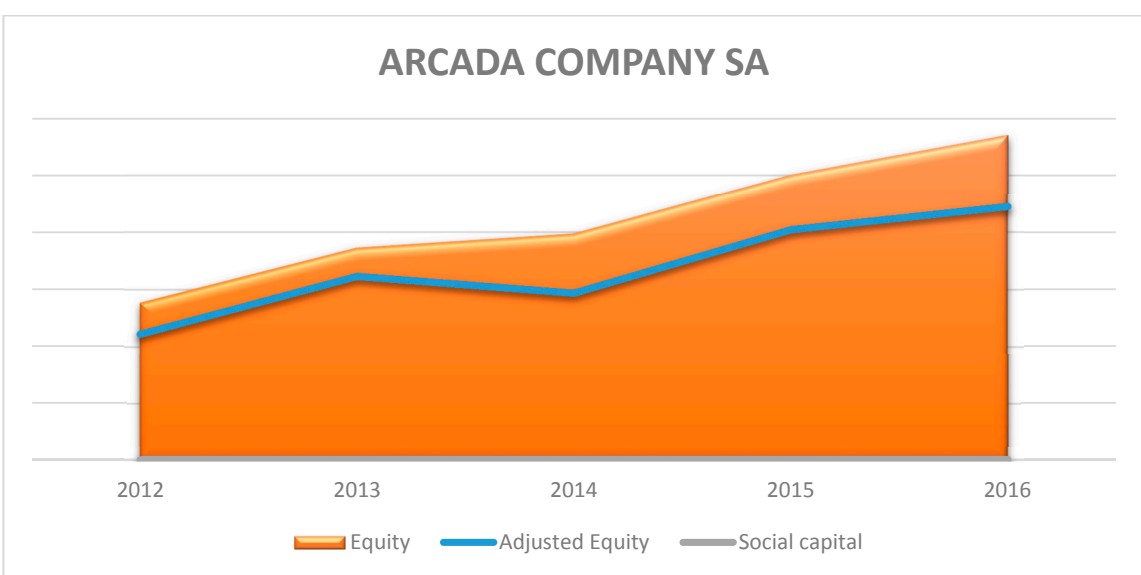

**Figure 9.** The increase in social capital following post balance sheet events adjustments for ARCADA COMPANY SA GALATI.

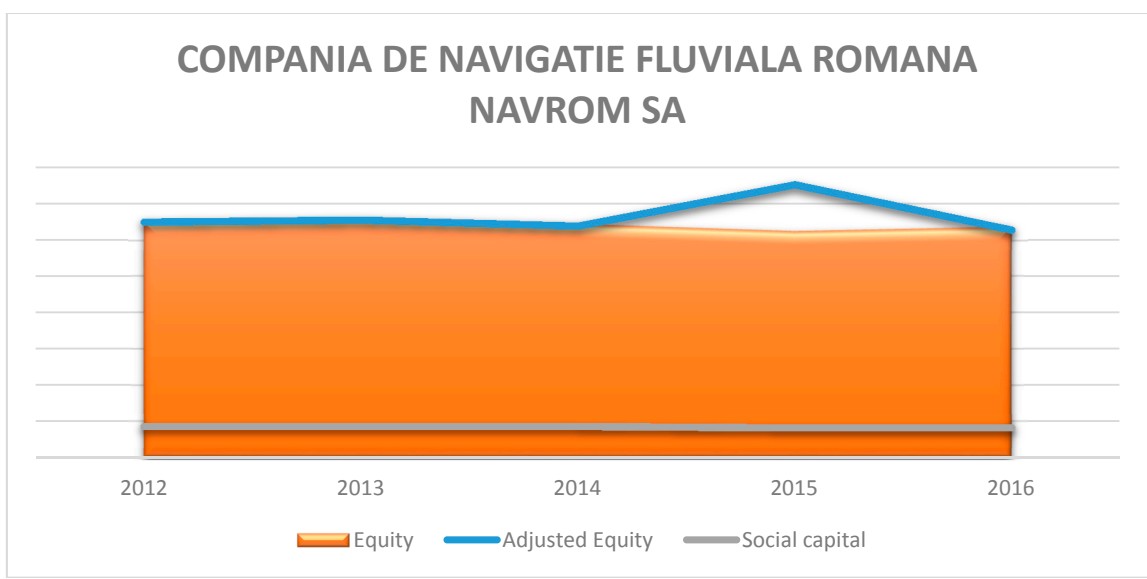

**Figure 10.** The increase in social capital following post balance sheet events adjustments for COMPANIA DE NAVIGATIE FLUVIALA ROMANA NAVROM SA GALATI.

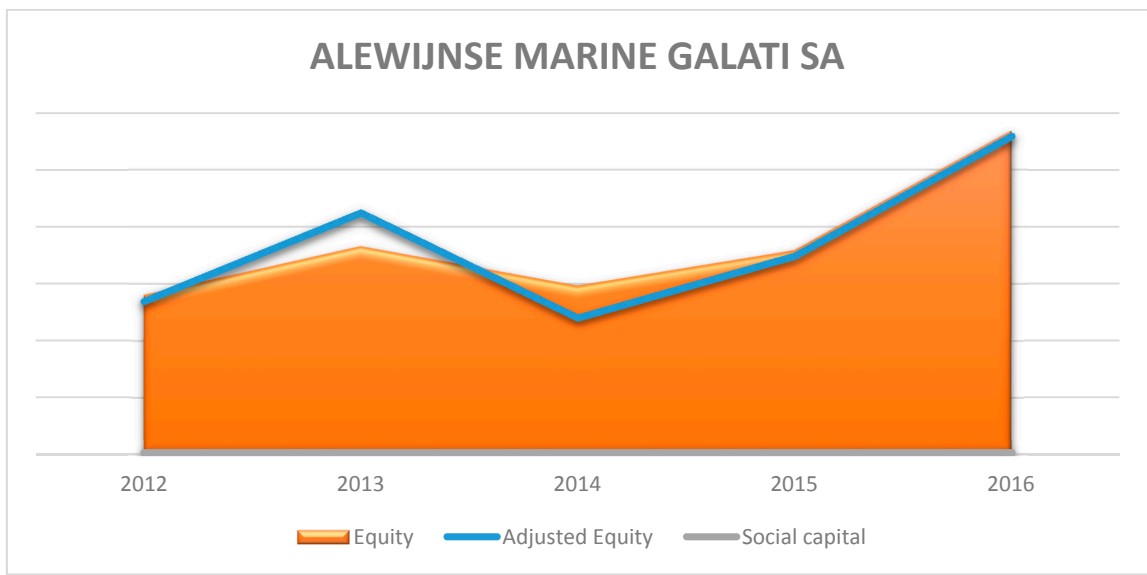

**Figure 11.** The increase in social capital following post balance sheet events adjustments for 5. ALEWIJNSE MARINE SA GALATI.

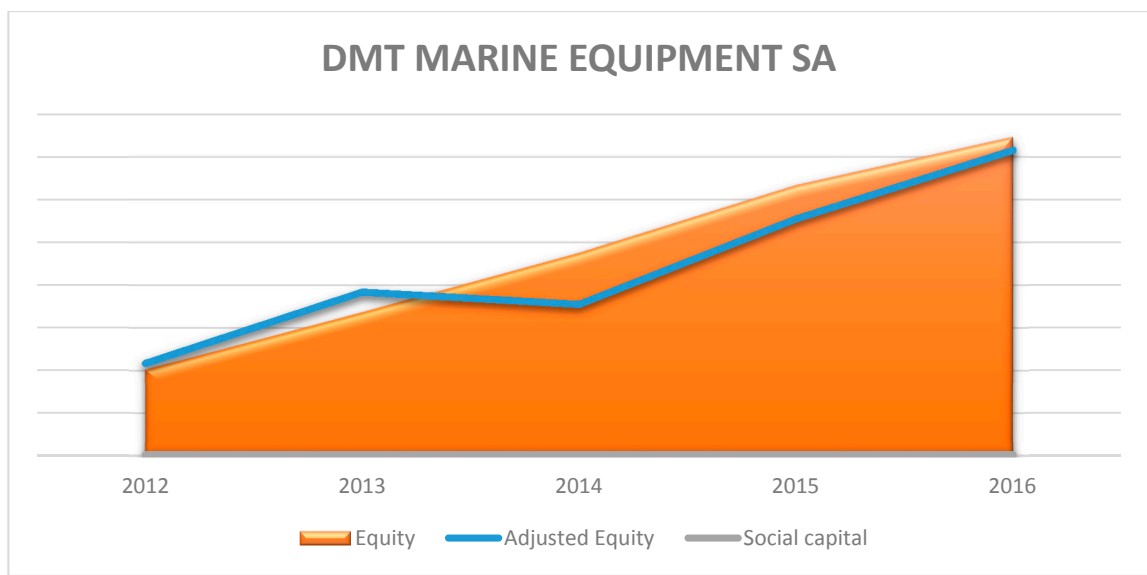

**Figure 12.** The increase in social capital following post balance sheet events adjustments for DMT MARINE EQUIPMENT SA GALATI.

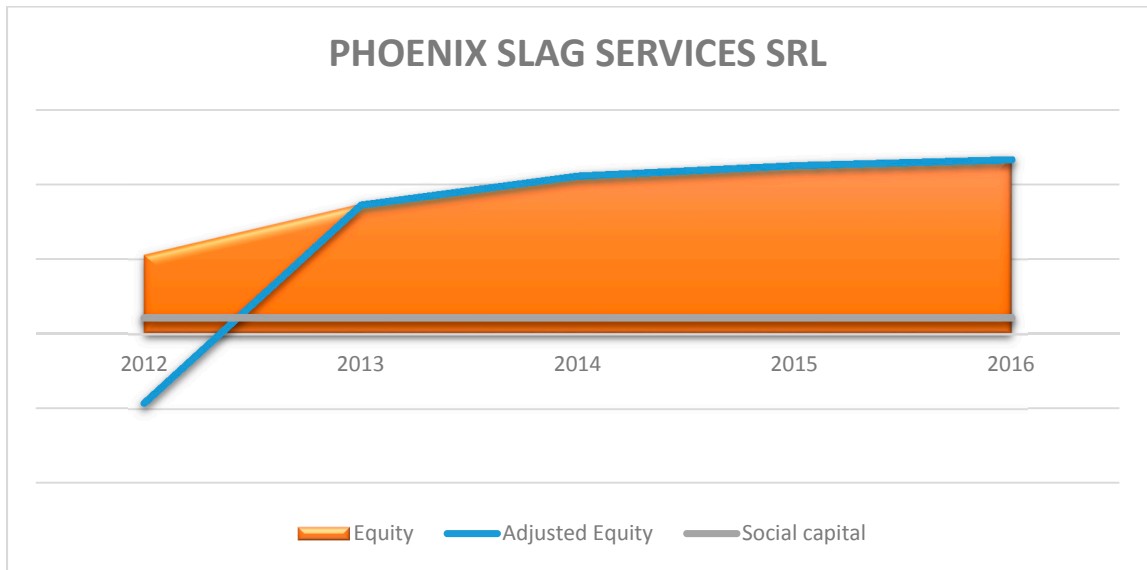

**Figure 13.** The increase in social capital following post balance sheet events adjustments for PHOENIX SLAG SERVICES SRL.

There is an increase in the analyzed shareholders' equity stake, which, following the use of the model, proved to be realistic, except for 2012, the Equity and Adjusted Equity trend equations being approximately equal for 2013–2016. The increase is based on negative reserve adjustments for 2012, which generated an inflexion point and positive provisions and net income adjustments. Thus, at the end of 2016, the company accounted for a goodwill based on the positive net result and an increase in fixed assets and current assets by 108% compared to 2011 (Figure 14).

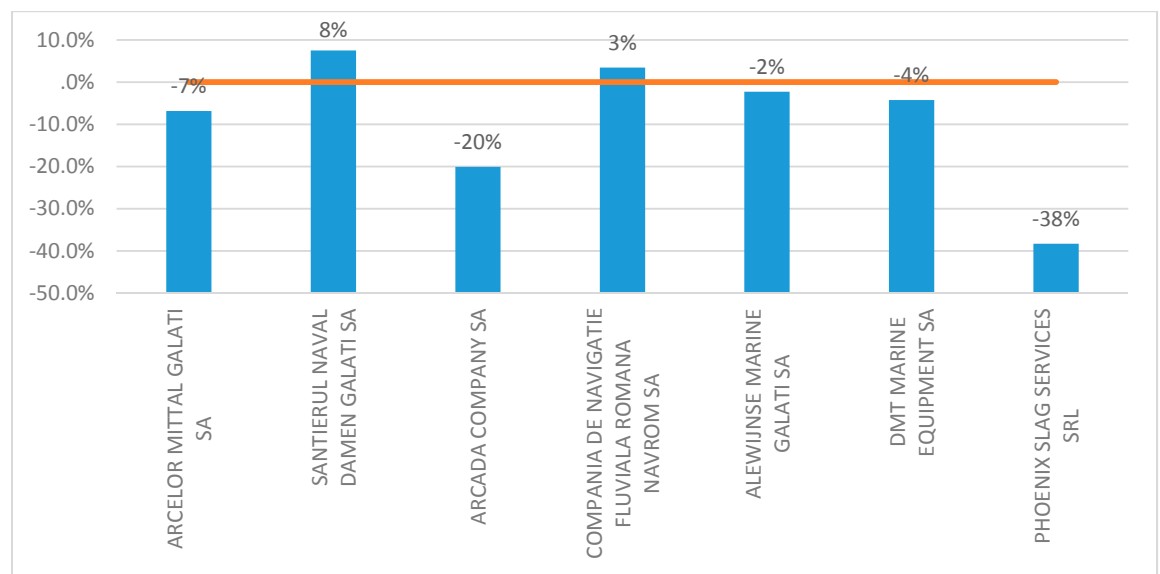

**Figure 14.** Equity adjustments average range based on the proposed model.

## 4. Discussion

Regarding the working hypotheses, we performed the testing of the results obtained by applying the model using the econometric modeling statistical procedures—regression models, Ordinary least squares (OLS) and Two—stage least squares (TSLS), obtaining statistically significant thresholds (H2, H3) for the time series 2012–2016, with 2011 being excluded due to dynamic values of the data series. The Impact on reported equity and impact of social capital increase/decrease are shown in Table 5.

**Table 5.** Econometric models for testing hypothesis H2 and H3.

| TSLS | Dependent Variable | Instrumented | Instruments | Coefficient | Std. Error | *t*-Ratio | *p*-Value | |
|------|--------------------|--------------|-------------|-------------|------------|-----------|-----------|---|
| 2012 | Social capital | Equity | Adjusted Equity | 166.325 | 0.293587 | 5.665 | 0.0013 | *** |
| 2013 | Social capital | Equity | Adjusted Equity | 145.559 | 0.201182 | 7.235 | 0.0004 | *** |
| 2014 | Social capital | Equity | Adjusted Equity | 127.770 | 0.131042 | 9.750 | <0.0001 | *** |
| 2015 | Social capital | Equity | Adjusted Equity | 108.931 | 0.0936929 | 11.63 | <0.0001 | *** |
| 2016 | Social capital | Equity | Adjusted Equity | 0.817073 | 0.0416343 | 19.62 | <0.0001 | *** |

*** High statistically significant threshold.

The H1 hypothesis in relation to the H4 hypothesis was also tested by econometric modeling using the Ordinary least squares (OLS) method, with statistically significant data on the impact of the model on the equity values (Table 6).

**Table 6.** Econometric models for testing hypothesis H1 and H4.

| OLS Instrumented | Coefficient | Std. Error | *t*-Ratio | *p*-Value | |
|------------------|-------------|------------|-----------|-----------|---|
| Impact on reported equity | 436.422 | 874.205 | 4.992 | 0.0379 | ** |
| Impact on reported equity | 189.241 | 513.594 | 3.685 | 0.0664 | * |
| Impact on reported equity | −50.9935 | 759.114 | −6.718 | 0.0215 | ** |
| Impact on reported equity | −20.4495 | 783.792 | −2.609 | 0.1208 | |
| Impact on reported equity | −3.73293 | 0.573162 | −6.513 | 0.0228 | ** |

**, * High statistically significant threshold.

The statistical tests performed showed that the OLS estimates are consistent for the 6 econometric models elaborated through the Gretl statistical program (values obtained by the Hausman test) as well as the statistical representativeness value for the 6 models, it is framed in the incidence range of

90–100% (R-squared test), which demonstrates that all the assumptions have been confirmed and the proposed model is valid and representative of the studied phenomenon.

Compared with other models studied (Advani and Malde 2018), the model proposed by the authors represents a quantitative measurement of the impact of the errors on the accounting information and their correction based on the analysis of the events after the balance sheet. Advani and Malde have identified network models that evaluate network entropy based on the variables used to identify errors. Unlike the model presented above, the authors of the study highlight the magnitude of errors based on the quantitative models and data reported by the entities examined and manage to identify their impact on the reported own capital. Another model that evaluates errors in terms of damage (Fang et al. 2017), analyzes the probability of interfering with errors in the accounting reporting system, taking into account the interest of managers and the incentives offered to them. The proposed model is an econometric regression type in which the damage is the dependent variable, and the degree of error is matrix-matched to the regression coefficients of the variables. Compared with this model, which has a probabilistic character and is adjusted by ambiguous quarks, our model adds value. This added value materializes by statistically analyzing the impact of errors on the reported equity. The quantifiable regression coefficients are determined on certain data. The significance threshold of the standard variable of adjusted equity is the maximum in all the analyzed cases. Xing and Yan (Xing and Yan 2019) have developed an econometric model of multiplicative type in which the quality of the information system reduces the systemic risk. The model contains certain variables (size, ROA, leverage, net capital expenditures, R&D) to which a number of qualitative variables are added (market–to-book, sales Herfindhal, business segments, constant, observations) which reduce the validity of the model through their subjective assessment side. Both the model presented by Xing and Yan, as well as our proposed model, addresses systemic risk through retrospective treatment and evaluate different quality of accounting information by different procedures. From the perspective of the differences between the two models, we can notice that our model has the more applied character, the elimination of the qualitative variables and the validation of the results on certain data. The model proposed by Zakolyukina (Zakolyukina 2018) represents an originality in the dynamic approach of systemic risk produced in the interest of changing the accounting values. Considering that the model was reported to GAAP (Zakolyukina 2018), the model proposed by us has few elements that can be correlated with the model in question. It is highlighted that the errors evaluated by our model can be integrated into the model of the researcher, bringing value to the dynamic model proposed by it.

## 5. Conclusions

The analysis of the model based on average values for the Adjusted Equity/Equity ratio shows that most companies account for figures below the economic reality at the balance sheet date, which require negative equity adjustments. Companies operating in the naval, shipbuilding and shipping industries are the only ones reporting equity values above the real market value, with this approach being favored by the specific evolution of the field.

The proposed model helps to narrow the differences in terms of the reality of the market in which the business runs and to reduce the risk of reporting errors that may pose future tax risks.

At the same time, from the shareholders/associates' perspectives, the model makes it possible for the size of the property owned in terms of assessed equity to be updated to its real value, as well as for the interests stakes to be readjusted.

The proposed model is of real interest to managers due to the rapidity of the transformation of accounting information and its ability to predict future trends based on the historical data taken from the financial statements. This model is suited to an IT integration in ERP (Enterprise Resource Planning) and accounting models for business entities.

**Author Contributions:** All authors contributed equally to this work.

**Funding:** This research received no external funding.

**Conflicts of Interest:** All authors declare no conflict of interest.

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
