# Peer review of "Correction of Accounting Errors through Post Balance Sheet Event Analysis for Romanian Companies"

_economies, doi:10.3390/economies7020029_

Round 1

Reviewer 1 Report

The manuscript must be thoroughly revised in terms of writing and editing. Your research must be properly integrated in the current WoS literature and substantially related to other case studies.

Author Response

Comments and Suggestions for Authors

The manuscript must be thoroughly revised in terms of writing and editing. Your research must be properly integrated in the current WoS literature and substantially related to other case studies.

Answer: We reviewed the articulation in terms of native English. We have introduced new bibliographic references with reference to current case studies on the chosen topic (6 references) and we compared the research results with the other research models in the field (please see lines 165-188).

We reviewed the quoted sources and we have made changes to the citations in the text to fulfill your requirements. We have supplemented our research with 20 other companies (see Table 1, 211 line) by increasing the study sample to a total of 40 companies.

We have completed the discussion section by connections to other case studies (please see lines 482-511).

We hope our efforts fulfill your requirements altogether.

Thank you for your suggestions.

Reviewer 2 Report

The subject of the research is not easily understood. It is stated that the purpose is to reveal the impact on the quality of reported financial data. The introduction presents a review of the accounting estimates and the model presented in the content is based on the evolution of some indicators that can be determined on the basis of the information in the balance sheet and which are embedded in a risk model. What is the correlation with accounting errors? It is recommended to specify such elements of correlation.

The author uses accounting terms that are not specific to Romanian regulations (for example: provisions for income in advance). It is recommended to review this aspect.

The author refers to a series of indicators that are not used in Romanian practice and which should be explained in terms of their usefulness (for example: expenses in advance/equity). The analysis does not show what the parameter ¯1,6 represents. Additional specifications are recommended.

The author refers to the debt / profit indicator, but, for example, ARCELOR MITTAL from 2009 to 2017 recorded losses. What does the calculated indicator mean? Does it assimilate the loss of the net profit realized? It is recommended to specify these aspects.

The author uses the phrase ”IAS 8 standard”, that is a pleonastic reference. Reformulation is recommended.

Author Response

Comments and Suggestions for Authors

The subject of the research is not easily understood. It is stated that the purpose is to reveal the impact on the quality of reported financial data. The introduction presents a review of the accounting estimates and the model presented in the content is based on the evolution of some indicators that can be determined on the basis of the information in the balance sheet and which are embedded in a risk model. What is the correlation with accounting errors? It is recommended to specify such elements of correlation.

Answer: We have introduced new studies (6) presented in the current literature on the correction of errors and we descused about how they are addressed in relation to systemic risk (please see lines 165-188) and the quality of accounting information.

We reviewed the sources quoted in order to eliminate the possibility of their presentation in the text.

 We have completed the introduction regarding the assessment of the effect of accounting policy changes and the application of retrospective treatments (please see lines 48-75). We have added cooments about the impact of IAS 10 on the phenomenon studied (please see lines 80-100).

 We added a motivation at the end of the introduction about the purpose of the research and the correlation between accounting errors and the risk of post-balance sheet events (please see lines 189-198).

The author uses accounting terms that are not specific to Romanian regulations (for example: provisions for income in advance). It is recommended to review this aspect.

Answer: We have removed the non-specific "Provisional Advance Income" and modified the risk treatment by referring to changing the destination of the amount, in "pending amounts" (please see line 360).

The author refers to a series of indicators that are not used in Romanian practice and which should be explained in terms of their usefulness (for example: expenses in advance/equity). The analysis does not show what the parameter ¯1,6 represents. Additional specifications are recommended.

Answer: Parameter 1-6 represents the number of years and the interpretation of this parameter was explained at line 211.

The "expenses in advance/equity" indicator mentioned is defined starting with line 285 in the article (please see lines 285-290), adjusting the provision referred to row 346 (advance expenditure/equity) should be interpreted in correlation with the defined indicator starting with line 285.

The utility of the indicator resides in the probable risk through the risk profile defined in line 346 and the following one, under the conditions of an average impact assessed at the firm level according to the calculations shown in Table 4 (where dynamics indicator is presented) and according to Table 5 where adjustments are made. Practically, without the breakdown by type of indicators, the adjusted values can’t be properly calculated in Table 5.

The author refers to the debt / profit indicator, but, for example, ARCELOR MITTAL from 2009 to 2017 recorded losses. What does the calculated indicator mean? Does it assimilate the loss of the net profit realized? It is recommended to specify these aspects.

Answer: We have modified the indicator that was initially defined as "Debt / Net Profit or Net Losses" (please see line 371, Table 3).

The author uses the phrase ”IAS 8 standard”, that is a pleonastic reference. Reformulation is recommended.

Answer: We have reviewed the whole article and removed the pleonastic reference.

We hope our efforts fulfill your requirements altogether.

Thank you for your suggestions.

Reviewer 3 Report

The paper deals with the correction of accounting errors through post balance sheet in selected companies. The article is formally successful. 

I consider the literature review to be inadequate. I recommend that the literature review should be elaborated, supplemented by the findings of the authors, etc. 

Results include only a description of the resulting values without a more detailed explanation of what the values mean and what the resulting values mean for the enterprise and discussion is missing.

Additionally, authors rely on the general use of the model based on the analysis of a small number of entities - only 20 companies.

Author Response

Comments and Suggestions for Authors

The paper deals with the correction of accounting errors through post balance sheet in selected companies. The article is formally successful. 

Answer: Thank you for your appreciation.

I consider the literature review to be inadequate. I recommend that the literature review should be elaborated, supplemented by the findings of the authors, etc. 

Answer: We have introduced new studies (6) presented in the current literature on the correction of errors and we descused about how they are addressed in relation to systemic risk (please see lines 165-188) and the quality of accounting information.

We have completed the introduction regarding the assessment of the effect of accounting policy changes and the application of retrospective treatments (please see lines 48-75). We have added cooments about the impact of IAS 10 on the phenomenon studied (please see lines 80-100).

 We added a motivation at the end of the introduction about the purpose of the research and the correlation between accounting errors and the risk of post-balance sheet events (please see lines 189-198).

Results include only a description of the resulting values without a more detailed explanation of what the values mean and what the resulting values mean for the enterprise and discussion is missing.

Answer: We have introduced a new chapter of discussions (chapter 4) restructuring the old chapter results and discussions. All calculated values were transposed by figures 6 to 12 in case studies per company.

Case studies have generally highlighted a difference between equity and equity values, adjusted for the application stages of provisions, resulting in slightly adjusted asymptomatic curve behavior of unadjusted values.

 All interpretations of the specific aspects of the 7 companies were entered in the resulting chapter starting with line 390, up to line 454 inclusive.

Figure 13 shows the behavior for each firm in relation to the moment 0 set by model (neutral point). As ia detailed at line 477, statistical tests for smallest squares (OLS) or smallest squares method (TSLS) reveals  the significant impact of the model (line 471 and 476). Additionaly, we completed the discussion section by referring to other case studies (please see lines 482-511) and we have highlight the personalized characteristics of the proposed model in relation to other models published lately (2017-2018) in the literature.

Additionally, authors rely on the general use of the model based on the analysis of a small number of entities - only 20 companies.

Answer: We have supplemented our research with 20 other companies (please see Table 1, line 211) by increasing the study sample to a total of 40 companies.

We hope our efforts fulfill your requirements altogether.

Thank you for your suggestions.

Round 2

Reviewer 1 Report

The manuscript has been poorly revised and the quality of language is even lower. Both the abstract and text start strangely. The text in figure 1 is not in English. The content of the manuscript is textbook-like in generality.

Reviewer 2 Report

The author has reviewed the work in accordance with the reviewer's comments.